# NopC/T/L Signal Crosstalk Gene *GmPHT1-4*

**DOI:** 10.3390/ijms242216521

**Published:** 2023-11-20

**Authors:** Zikun Zhu, Tong Yu, Fuxin Li, Yu Zhang, Chunyan Liu, Qingshan Chen, Dawei Xin

**Affiliations:** National Key Laboratory of Smart Farm Technology and System, Key Laboratory of Soybean Biology in Chinese Ministry of Education, College of Agriculture, Northeast Agricultural University, Harbin 150030, China; zzk13009707019@163.com (Z.Z.); yutong6488@163.com (T.Y.); lifuxin202310@163.com (F.L.); zhangyu_05101999@163.com (Y.Z.); cyliucn@126.com (C.L.)

**Keywords:** signal crosstalk, *SinoRhizobium fredii* HH103, HH103Ω*NopT&NopC&NopL*, *GmPHT1-4*

## Abstract

Symbiotic nodulation between leguminous plants and rhizobia is a critical biological interaction. The type III secretion system (T3SS) employed by rhizobia manipulates the host’s nodulation signaling, analogous to mechanisms used by certain bacterial pathogens for effector protein delivery into host cells. This investigation explores the interactive signaling among type III effectors HH103Ω*NopC*, HH103Ω*NopT*, and HH103Ω*NopL* from *SinoRhizobium fredii* HH103. Experimental results revealed that these effectors positively regulate nodule formation. Transcriptomic analysis pinpointed *GmPHT1-4* as the key gene facilitating this effector-mediated signaling. Overexpression of *GmPHT1-4* enhances nodulation, indicating a dual function in nodulation and phosphorus homeostasis. This research elucidates the intricate regulatory network governing *Rhizobium*–soybean (*Glycine max* (L.) Merr) interactions and the complex interplay between type III effectors.

## 1. Introduction

Soybean (*Glycine max*) serves as a pivotal crop globally, providing a sustainable and cost-effective plant-based protein alternative to meat [1]. The application of nitrogen during R1 or the early flowering stage has been correlated with increased soybean yields [2]. However, the overuse of nitrogenous fertilizers has heightened environmental concerns, necessitating a shift towards biological nitrogen fixation methods [3,4].

Symbiotic interactions between soybeans and specific bacteria result in the formation of nodules, specialized organs for biological nitrogen fixation. The *BradyRhizobium elkanii* USDA61 strain exhibits a functional T3SS, pivotal for host–legume symbiosis specificity [5]. Contrastingly, *BradyRhizobium japonicum* is noted for its saprophytic abilities and competitive edge, while *BradyRhizobium diazoefficiens* is recognized for its superior nitrogen fixation efficacy [6]. The symbiotic relationship is critical for enhancing nitrogen assimilation by soybean roots, converting atmospheric nitrogen into a usable form, ammonia, thus augmenting soybean yield [7]. T3SS facilitates the secretion of virulence factors, also termed type III effector factors, which are integral to both nodule development and modulation of the plant host’s immune response [8]. Investigating the role of these effectors is essential to optimize soybean yields independently of synthetic nitrogen inputs.

Rhizobia, which are Gram-negative bacteria, possess T3SS, a sophisticated structure that is distinctive to pathogenic Gram-negative bacteria. This system allows the direct translocation of effector proteins into the host cell cytoplasm, circumventing the extracellular milieu [9]. The suite of type III effector factors secreted by *Rhizobium*’s T3SS, including NopA, NopB, NopC, NopT, NopD, NopP, NopZ, NopL, NopM, NopX, and NopJ, along with transcriptional activator TtsI, have been identified and play a pivotal role in the nodulation process [10,11,12,13,14,15].

The expression of NopC is contingent upon the presence of flavonoids and the regulatory activity of transcriptional regulators NodD and TtsI. Notably, NopC does not have analogous counterparts in bacterial pathogens and is devoid of conserved domains or characteristics that might elucidate its symbiotic function. It has been hypothesized that NopC may function as a chaperone, aiding in the assembly of secretion machinery or the efflux of effectors into the host cell [16]. NopC secretion relies on T3SS and is introduced directly into the root cells of the soybean via this system [17,18]. Typically, T3SS-associated proteins are acidic, cytoplasmic, and retained within the bacterial cell, with some encoded by operon genes linked to secretion apparatus components. Interestingly, inactivation of NopC does not abolish the secretion of other Nops. Among the effectors studied, NopC appears to be a principal factor in obstructing nodulation in both *Rhizobium* and *Lotus japonicus Gifu* [19]. NopC mutants retain the ability to infect *L. japonicus Gifu* through infection threads rather than intercellular invasion, indicating a role in shifting the mode of infection [20].

Similarly, NopL is a rhizobia-specific protein without parallels in bacterial pathogens. Research has demonstrated that NopL mitigates premature nodule senescence by disrupting host cell MAPK signaling pathways [21,22]. Nodules induced by the NGR234Ω*NopL* mutant displayed an increased incidence of black necrotic spots, indicative of impaired nodule formation, compared to NGR234 inoculation [23]. Quantitative analysis revealed a greater frequency and accelerated progression of nodule necrosis post-inoculation with the NGR234Ω*NopL* mutant, underscoring the detrimental impact of the NopL effector [24]. While NopL is not essential for Nop secretion, it is necessary for effective nodulation in certain host species, suggesting that its role as an effector protein is host-genotype dependent [25]. Additionally, NopL can be multiphosphorylated by MAP kinases, indicating a potential regulatory interaction [22]. These findings affirm the significance of type III effectors in nodulation control.

NopT, a member of the YopT/AvrPphB family, shares high homology with YopT of *Yersinia*, LopT of *Photorhabdus luminescens*, and AvrPphB of *Pseudomonas syringae* [26,27]. This effector is active on isopentenylated GTPases [28]. It interacts with ATP-CSACP2 (ATP-citrate synthase alpha chain protein 2), HIRP (hypersensitive-induced response protein), and proteins of *Robinia pseudoacacia* to modulate the immune response during rhizobial infection. Analogous to NopL, NopT’s transient expression in *Nicotiana benthamiana* results in leaf necrosis and elicits a hypersensitive response (HR) [29]. Additionally, NopT induction of *GmTNRP1*, an LRR-RK family protein at the cell membrane, serves to downregulate nitrogenase activity [30].

The signaling network among type III effector proteins is complex, exhibiting crosstalk, and certain transporters are pivotal in signal mediation.

In *Arabidopsis thaliana*, the *PHT1* family comprises nine phosphorus transporters. *AtPHT1-1* and *AtPHT1-4* function as high-affinity phosphate transporters under varying phosphorus levels, with their expression being upregulated under conditions of phosphorus deficiency [31] and exhibiting high transcription in the roots [32]. In rice (*Oryza sativa*), *OsPht1*, part of the *Pht8* family, regulates phosphate homeostasis [33]. The PHR-PHT1 module is integral to Pi stability, and high expression levels of *GmPHT1* in root nodules contribute to increased Pi accumulation and nodule growth, in addition to enhancing nitroxidase activity [34,35]. *GmPHT1-4*, a member of the PHT family with 14 homologous genes, is highly expressed in the roots, similar to *AtPHT1* in *Arabidopsis*, playing a critical role in phosphate uptake under both low and high phosphate conditions [36].

The type III secretion system (T3SS) facilitates the transfer of a suite of effector proteins with diverse functions directly into the host cell. Research indicates that double mutants of type III effectors can elucidate synergistic relationships among them. It is posited that a host gene may concurrently respond to multiple type III effectors and that the host signaling network might be modulated by a specific hub gene during symbiosis establishment. To identify soybean genes responsive to NopC, NopT, and NopL, RNA-seq analysis was conducted to discern differentially expressed genes (DEGs) following inoculation with single and triple mutants of these effectors. Subsequently, we screened these candidate genes to characterize their functions at the transcript level.

## 2. Results

### 2.1. HH103ΩNopT&NopC&NopL Construction of Mutant

Building upon the previously established signal crosstalk between NopT and NopP, which suggested an interconnection between these nodulation outer proteins [37], subsequent investigations have delineated a shared signaling network involving NopC, NopT, and NopL. In this context, we generated the HH103Ω*NopT&NopC&NopL* mutant strain, hereafter referred to as TCL, through triparental mating of NopT with the HH103Ω*NopL&NopC* strains. This was achieved by employing Overlap PCR to construct a 3030 bp NopT-Cm fragment. First, the desired fragment was obtained by PCR reaction (Appendix A), which was subsequently integrated into recombinant suicide vector pJQ200SK. Confirmation of the correct assembly was obtained by PCR, which yielded the expected band size of 3030 bp (Appendix A). Further validation of the TCL mutant was conducted through both PCR and Southern blot assays, using the primer sets NopT-S-F-X, NopT-A-R-X, Cm-F, and Cm-R. The primers NopT-S-F-X and NopT-A-R-X facilitated detection of the NopT-Cm fragment, whereas the Cm-F and Cm-R primers verified the presence of a 960 bp antibiotic resistance segment. Additionally, the primer pair NopT-S-F-X and Cm-R enabled identification of both the antibiotic resistance gene and the upstream NopT fragment, which collectively measured 2020 bp. This concordance was evidenced by positive identification of the TCL strain in lane 6 during agarose gel electrophoresis (Figure 1). Comparative genomic analysis of TCL and wild-type *Rhizobium* HH103, post-*Xho* I restriction endonuclease digestion, revealed fragments measuring approximately 3000 bp and 2000 bp, respectively (Appendix A). These findings conclusively demonstrated successful construction of the TCL mutant.

### 2.2. TCL Positively Regulates Nodules

To investigate the impact of various mutants on nodulation, DN50 soybean plants were inoculated with wild-type strain HH103, individual mutants NopC, NopT, NopL, and TtsI, and the triple mutant TCL. The nodule counts and their dry weights were subjected to statistical evaluation. When compared to wild-type HH103, a notable decrease in nodulation was observed in the mutants TtsI, TCL, NopC, NopT, and NopL. Notably, among the single mutants, the NopC mutant exhibited the most pronounced reduction in nodulation rate (Figure 2). The triple mutant TCL, along with the NopC, NopT, and NopL mutants, did not demonstrate statistically significant reductions in nodule count, suggesting that simultaneous mutations of NopC, NopT, and NopL did not exacerbate the reduction in nodule number (Figure 2b, Appendix A).

Further analysis of the nodule dry weights indicated that nodules from plants inoculated with the TCL mutant had greater mass, a trend that was also present, albeit to a lesser extent, in nodules from plants inoculated with the NopC, NopT, and NopL mutants (Figure 2c, Appendix A). Therefore, it could be deduced that the type III effectors NopC, NopT, and NopL play a positive regulatory role in nodule formation, and there exists a synergistic signal interaction among these effectors.

### 2.3. Identification of DEGs Induced by NopC, NopT, NopL, and TCL

The entry of type III effectors into plant cells triggers a specific genetic response that influences nodulation. To delineate the gene networks responsive to the type III effectors NopC, NopL, and NopT, RNA-seq was employed. Differential expression analysis was conducted at 0.5 h post-inoculation (hpi) and 6 hpi with mutant strains NopC, NopL, and NopT. Comparative analysis utilized volcano plots to compare the DEGs at both time points against a control setup involving a mock treatment with MgSO_4_ and wild-type *Rhizobium* HH103 inoculation (Appendix A).

The analysis indicated that at 0.5 hpi, inoculation with NopC resulted in 2235 DEGs in comparison to the HH103 control. This number increased to 2364 DEGs at 6 hpi. For NopL, 1762 DEGs were observed at 0.5 hpi, but this number significantly declined to 1190 DEGs after 6 hpi. Inoculation with NopT elicited 1736 DEGs at 0.5 hpi, with a substantial reduction to 870 DEGs at 6 hpi.

A comparison between the triple mutant TCL and HH103 at 0.5 hpi revealed 905 DEGs, including 97 upregulated and 775 downregulated genes. At 6 hpi, the total DEGs diminished markedly to 338, with 45 genes being upregulated and 298 downregulated (Figure 3). These findings highlighted a substantial number of DEGs affected by NopC, NopT, NopL, and TCL, implying the existence of a redundant gene regulatory network among the three Nop effectors.

### 2.4. Weighted Gene Correlation Network Analysis (WGCNA)

Weighted Gene Correlation Network Analysis (WGCNA) was applied to a selection of 2290 genes exhibiting an average fragment count per million mapped reads (FPKM) greater than 1 at 0.5 hpi and 860 genes at 6 hpi in plants inoculated with NopC, NopT, NopL, and TCL. A heatmap representing the co-expression network was generated based on the correlation coefficients of these genes (Figure 4 and Figure 5), with modules reflecting highly correlated gene clusters delineated in this heatmap (Figure 4a and Figure 5a).

At 0.5 hpi, three primary gene clusters were identified, distinguished by color coding. The turquoise module showed the strongest correlation with TCL inoculation and contained a majority of upregulated genes (Figure 4d). At 6 hpi, eight major gene clusters were identified, with the red module displaying the highest correlation with TCL, again containing predominantly upregulated genes (Figure 5d). Clusters with significant correlations were further analyzed to construct Kyoto Encyclopedia of Genes and Genomes (KEGG) and Gene Ontology (GO) enrichment maps (Figure 6 and Figure 7).

For the module selected at 0.5 hpi, 1412 genes were associated with TCL and were notably enriched in the plant–microbe interaction and phytohormone signaling pathways. The module at 6 hpi comprised 67 genes associated with TCL, predominantly enriched in the phytohormone signal transduction pathway. A distinct enrichment in isoflavone biosynthesis was observed at 6 hpi compared to 0.5 hpi, indicative of *Rhizobium*-triggered soybean nodulation initiation.

Subsequent screening of genes from the WGCNA, in conjunction with DEGs identified from Venn diagrams, led to the isolation of 16 TCL-associated genes at 0.5 hpi. These were mainly enriched in the phytohormone signaling and plant–pathogen interaction pathways (Figure 5b). At 6 hpi, one TCL-associated gene was identified within the phosphorus transporter category, upregulated post-TCL inoculation. These 17 genes were proposed as potential network hubs for crosstalk among NopC, NopT, and NopL.

### 2.5. Screening for Signal Crosstalk Genes

Comprehensive annotation was conducted for 17 putative signal crosstalk genes (Table 1). Among them, *GmGPL1* (*Glyma.04G011900*), *GmPRX52* (*Glyma.06G145300*), *GmDUF1677* (*Glyma.10G247200*), and *GmPHT1-4* (*Glyma.10G036800*) demonstrated high expression levels in soybean rhizomes (Appendix A). Varied expression patterns were noted for other genes across different soybean tissues according to Phytozome data. Preliminary annotations and supporting literature posited *GmPHT1-4* as a pivotal gene in the signaling network. Concordance between RT-qPCR results and transcriptomic data for nodule-associated markers underpinned the transcriptome findings’ credibility (Figure 8).

### 2.6. Effect of OE-GmPHT1-4 on the Nodulation

The impact of *GmPHT1-4* on rhizome development was investigated using *GmPHT1-4*-overexpressing (OE) plants, developed through hairy root transformation (Appendix A). Soybean hairy roots harboring pSOY1-35S:*GmPHT1-4:GFP* constructs were grown in vermiculite for three days before exposure to HH103, NopC, NopT, NopL, and TCL. RT-qPCR analysis confirmed the overexpression of *GmPHT1-4* (Figure 9c). The nodule counts in OE-*GmPHT1-4* plants increased upon inoculation with HH103, NopC, NopT, and NopL, although the differences were not statistically significant (compared to the empty vector control; Figure 9a). Notably, OE-*GmPHT1-4* plants displayed a significant uptick in tumorigenesis post-TCL inoculation. Further assessment of the nodule dry weights indicated a substantial increase in OE-*GmPHT1-4* plants, with nodules also presenting increased size. Collectively, these findings suggested that *GmPHT1-4* augments rhizome growth, phosphorus homeostasis, and concurrently modulates nodule size.

### 2.7. Analysis of Difference in GmPHT1-4 Expression after Inoculation with Various Rhizobia

To elucidate the influence of various rhizobia on the expression of *GmPHT1-4*, RT-qPCR assays were conducted at 6 h post-inoculation (hpi) with strains HH103, NopC, NopL, NopT, TtsI, and TCL. The results indicated a pronounced elevation in *GmPHT1-4* expression in plants inoculated with rhizobial mutants in contrast to the wild-type strain. Notably, the amplified expression of *GmPHT1-4* correlated with an increment in nodule size, particularly post-TCL inoculation, where the expression surge of *GmPHT1-4* was most significant, and concomitantly, the nodules exhibited maximal enlargement. These findings corroborated the pivotal role of *GmPHT1-4* as a central gene within the integrated network of NopC, NopT, and NopL, pivotal for triggering signal transduction pathways in soybean nodulation.

## 3. Discussion

Type III effectors NopC, NopT, and NopL from *Rhizobium* are positively correlated with nodulation in soybeans [21,37]. Diminished expression of these effectors is associated with a significant decrease in nodule formation. Inoculation with the TCL mutant strain of *Rhizobium*, which encompassed combined mutations in NopC, NopT, and NopL, led to fewer nodules compared to wild-type *Rhizobium* HH103. Nonetheless, nodule numbers in TCL-inoculated plants did not significantly deviate from those in plants inoculated with single effector mutants, suggesting interactive effects among the type III effectors. RNA-seq analysis was employed to further investigate the intercommunication between NopC, NopT, and NopL. This analysis identified alterations in *GmPHT1-4* expression post-inoculation with varied rhizobia and following hairy root transformation, revealing the gene’s significant regulatory role in nodule formation (Figure 9 and Figure 10). This finding is consistent with prior reports. In rice, *OsPHT1-7* was shown to mediate the rapid accumulation of phosphorus in anthers during development, indicating its role as a specialized phosphorus transporter and function in anther development [38]. Analogously, in *Arabidopsis thaliana*, *Pht1-1* and *Pht1-4* are critical for phosphate uptake under both high and low phosphate conditions [32]. Although investigations into the response of *GmPHT1-4* under low phosphorus conditions have not been undertaken, it is posited that a similar response pattern would be observed in soybeans. The evident increase in *GmPHT1-4* expression following inoculation with *Rhizobium* mutants confirms its role within the downstream signaling pathways mediated by NopC, NopT, and NopL.

The initiation of legume–*Rhizobium* symbiosis is contingent upon intricate signaling cascades. Since 1999, forward and reverse genetic approaches have elucidated numerous genes related to nodulation, influencing both nodule development and function [39]. Transcriptomic analysis following inoculation with HH103 and its derivatives indicated elevated expression of *GmNIN2b*, *GmENOD40*, *GmNFR1*, and *GmNFR5*, albeit to different degrees. *GmNFR1* and *GmNFR5*, which are nodulation factor (NF) recognition receptors, have previously been linked to tumor formation [40]. Positioned downstream in the nodulation signaling pathway are *GmNIN2b* and *GmENOD40* [41]. The present study demonstrates that effector factors intersect with the NF signaling pathway, influencing its regulatory network. This research assessed the interplay between core genes like *GmPHT1-4* and nodulation signals, particularly in scenarios demanding recognition and signaling crosstalk. Predominantly, hub genes were associated with phytohormone signaling pathways. *GmPHT1-4* was notably enriched within the KEGG pathway as part of the MFS transporter protein pathway (K08176), indicating its involvement in phytohormone-mediated nodule organogenesis [40]. Furthermore, *GmPHT1-4*’s role in phosphorus homeostasis and rhizome size regulation suggests its potential linkage to polar cell growth. Previous studies have identified the Rho family of GTPases as pivotal regulators of eukaryotic polar cell growth, performing similar functions to the Small G Protein (SGP) class of signal transduction proteins [42]. Among the genes scrutinized, GLP-1, a glucose-1-phosphate adenylyltransferase, was implicated in starch biosynthesis through the generation of ADP-glucose from Glc-1-P and ATP. *GmPRX52* is classified within the peroxidase (POD) family, playing a role in the oxidative breakdown of indole acetic acid (IAA), thus influencing plant growth and morphogenesis [43]. Based on these findings, it is postulated that NopC, NopT, and NopL may similarly be involved in mechanisms governing polar cell growth.

Phosphorylation within plant cells can disrupt the MAPK signaling pathway, leading to the suppression of defense mechanisms [44]. NopT is capable of cleaving PBS1, thus altering nodule morphology by generating cleavage products with exposed terminal ends [37]. Moreover, NopL has been identified to contain multiple phosphorylation sites, with a prediction that both NopC and NopT also harbor such sites (Appendix A). It is plausible to consider that the activation of phosphorylation is essential for modulating nodulation by NopC, NopT, and NopL. Additionally, the inorganic phosphate transporter gene *GmPHT1-4* exhibits pronounced expression within nodules, signifying its contribution to nodule development. This suggests a mechanism whereby phosphorylation of NopC, NopT, and NopL may hinder the initiation of plant defense responses, thereby facilitating nodule formation in roots.

In summation, the present study corroborates the participation of *GmPHT1-4* within the signaling interplay among NopC, NopT, and NopL, specifically as a regulator of phosphorus (Pi) homeostasis subsequent to nodule formation. This regulatory function occurs downstream of *GmNFR1/5* and *GmNIN2b* signaling (Figure 11). The findings extend our comprehension of transporter signaling pertinent to the establishment of symbiosis between soybean and *Rhizobium*.

## 4. Materials and Methods

### 4.1. Acquisition of Triple Mutant HH103ΩNopT&NopC&NopL

A monoclonal strain of *Rhizobium* HH103Ω*NopL&NopC* was isolated on an agar plate and incubated at 28 °C until an OD_600_ of 0.6 was achieved. Concurrently, *Escherichia coli*, harboring the plasmids pJQ200SK-NopT-Spec and pRK2013, was cultured in liquid medium at 37 °C to an OD_600_ of approximately 0.6. The bacterial culture was then centrifuged in a 1.5 mL microcentrifuge tube at 8000 r/min for 5 min, after which the pellet was resuspended in 500 μL of antibiotic-free TY liquid medium. The bacterial suspension was prepared at a ratio of 2:1:1 for *Rhizobium*, helper, and recombinant suicide vector bacteria, respectively, followed by centrifugation at 8000 r/min for 5 min. After discarding the supernatant, the bacterial pellet was resuspended in 15 μL of TY liquid medium. Subsequently, 15 μL droplets of the bacterial mixture were plated on antibiotic-free TY solid medium and incubated at 28 °C for 36 h. Resulting large plaques were transferred to TY solid medium containing the corresponding antibiotic, a process that was repeated 2 to 3 times. Finally, monoclonal colonies were cultivated on TY solid medium with appropriate antibiotics and 5% sucrose to ensure the growth of monoclonal colonies, with the process repeated 2 to 3 times.

### 4.2. PCR Verification of Bacterial Solution

Mutant strain HH103Ω*NopT&NopC&NopL* served as the PCR template. The primers NopT-S-F-X, NopT-A-R-X, Cm-F, and Cm-R were employed for PCR to confirm the successful construction of the mutant strains, as indicated by the resultant band sizes post-electrophoresis.

### 4.3. Southern Blot

Probes for Southern blot analysis were designed with HH103Ω*NopT&NopC&NopL* as the template (Appendix A). The selected probe covered the junction between the NopT and Cm gene sequences, with an approximate length of 200 base pairs. The probe was hybridized to the DNA immobilized on a nylon membrane, with band detection achieved through radioautography.

### 4.4. Soybean Nodulation Experiment

The experimental subjects included two soybean (*G. max* L.) cultivars: Suinong 14 (SN14) and Dongnong 50 (DN50). DN50 is favored over SN14 for its higher efficacy in genetic transformation experiments. The bacterial strains utilized were Sino*Rhizobium* fredii and its derivatives: the single mutants HH103Ω*NopC*, HH103Ω*NopT*, HH103Ω*NopL*, HH103Ω*TtsI*, and the triple mutant HH103Ω*NopT&NopC&NopL*. Soybean cultivation conditions were maintained at 25 °C with a photoperiod of 16 h of light followed by 8 h of darkness. DN50 seeds, post-sterilization with chlorine, were sown in a double pot system. The plants were grown until the emergence of the first trifoliate leaf, at which point they were inoculated with the *Rhizobium* strains (Appendix A). Bacterial suspensions with an OD_600_ of 0.6 were prepared in 10 mM MgSO4. Each *Rhizobium* strain was inoculated onto 20 DN50 soybean plants at a volume of 1.5 mL per plant. Thirty days post-inoculation, 10 soybean plants exhibiting uniform growth were selected for the evaluation of nodulation phenotype.

B&D: MgSO_4_ 0.5 mol·L^−1^, Na_2_ MoO_4_ 0.2 mmol·L^−1^, MnSO_4_ 2 mmol·L^−1^, H_3_BO_3_ 4 mmol·L^−1^, CaCl_2_ 2 mol·L^−1^, CoSO_4_ 0.2 mmol·L^−1^, K_2_SO_4_ 0.5 mol·L^−1^, CuSO_4_ 4 mmol·L^−1^, KH_2_PO_4_ 1 mol·L^−1^, ZnSO_4_ 1 mmol·L^−1^, and C_6_ H_5_FeO_7_ 20 mmol·L^−1^.

### 4.5. Phenotypic Statistics and Data Analysis

Quantitative nodule counts were subjected to statistical analysis utilizing SPSS 22.0 for both *t*-tests and ANOVA. Histograms and box plots were generated using GraphPad Prism 8.0.1.

### 4.6. mRNA-Seq (mRNA Sequencing) Analysis

For transcriptomic analysis, the soybean cultivar SN14 was selected. Roots inoculated with MgSO_4_ (as controls), HH103, and the mutants HH103Ω*NopT*, HH103Ω*NopC*, HH103Ω*NopL*, and HH103Ω*NopT&NopC&NopL* were harvested at 0.5 h and 6 h post-inoculation. Root segments 1 cm in length were excised from near the hypocotyl for each of the three biological replicates, which weighed about 0.1 g. Transcriptomic data analysis was performed using DEseq2-edgeR, adopting an FDR of less than 0.01 and an FC greater than 1.5 as the thresholds. Heatmaps, KEGG pathway enrichment, GO annotations, and WGCNA were executed using TBtools-II v2.003.

### 4.7. RNA Extraction and RT-qPCR Analysis

Root samples were collected at 0.5 h and 6 h following rhizobial inoculation for RNA extraction using TRIzol Reagent (Thermo Fisher Scientific, Waltham, MA, USA) according to the supplier’s instructions. Complementary DNA (cDNA) was synthesized employing HiScript II Reverse Transcriptase, and RT-qPCR was conducted on a Roche LightCycler 480 System using TB Green (Takara Biomedical Technology, Japan). Each sample was represented by three biological and technical replicates, respectively. Ct values were used to calculate relative gene expression via the log2 (−ΔΔCt) method.

### 4.8. Hairy Root Transformation and Positive Soybean Root Detection

The full-length cDNA of *GmPHT1-4* was amplified from SN14-derived cDNA using RT-qPCR with *GmPHT1-4*-F/R primers (refer to Appendix A). The resultant construct, pSoy1-35S: *GmPHT1-4*: *GFP*, facilitated the generation of transgenic soybean hairy roots through Agrobacterium rhizogene-mediated transformation, following the protocol outlined by Kereszt et al. [45]. Selection of transgenic roots employed a LUYOR portable fluorescent protein excitation light source and subsequent verification by RT-qPCR. Transgenic plants were inoculated with HH103, its single mutants, and the triple mutant at an OD_600_ of 0.6, with 1.5 mL per plant. Nodulation counts and dry weight measurements were recorded 30 days post-inoculation (refer to Appendix A). These nodulation assessments were conducted across three independent experiments.

## Figures and Tables

**Figure 1 ijms-24-16521-f001:**
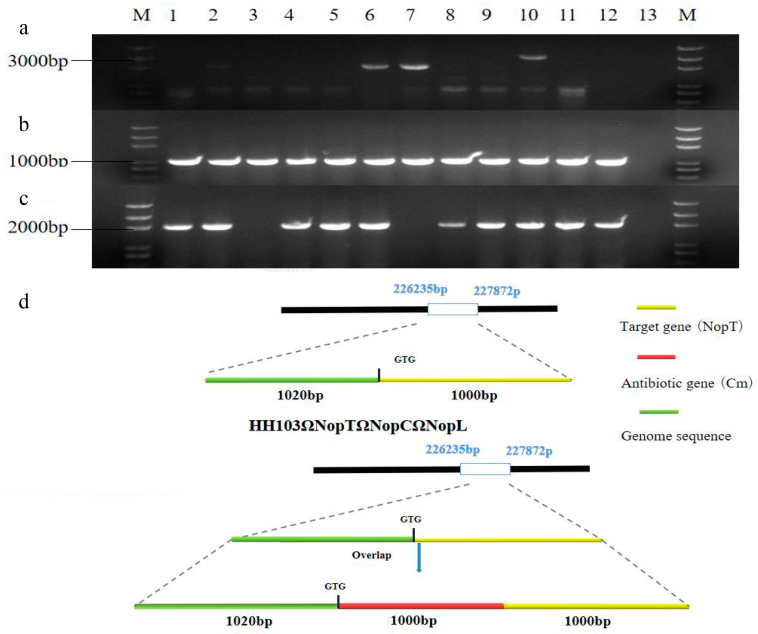
Construction and validation of mutant HH103Ω*NopC&NopT&NopL*. (**a**–**c**) Strain screening. The fragments in TCL are amplified by different primers. M: Trans 2K Plus DNA marker. (**a**) primers NopT-S-F-X and NopT-A-R-X; (**b**) primers NopT-S-F-X and Cm-R; (**c**) primers Cm-F and Cm-R. (**d**) HH103, NopT structure diagram and HH103Ω*NopC&NopT&NopL*, *NopT*-*Cm* structure diagram.

**Figure 2 ijms-24-16521-f002:**
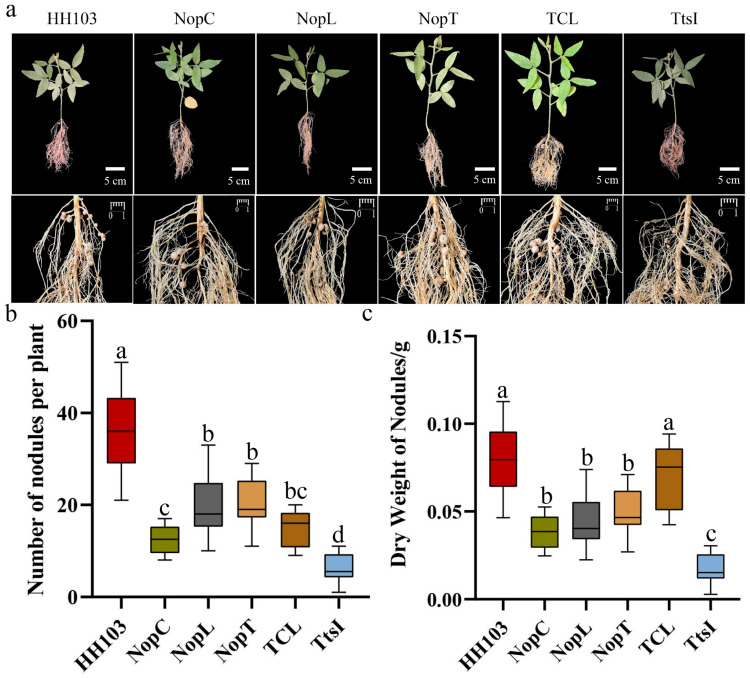
Effects of inoculation with wild-type *Rhizobium* HH103, single mutants NopC, NopL, NopT, and TtsI, and triple mutant TCL on DN50 soybean nodules. (**a**) The phenotypes were inoculated with wild-type HH103, NopC single mutant (HH103Ω*NopC*), NopT single mutant (HH103Ω*NopT*), NopL single mutant (HH103Ω*NopL*), and NopC, NopT, and NopL triple mutant (HH103Ω*NopT&NopC&NopL*), respectively. (**b**,**c**) After inoculation with wild-type HH103, NopC single mutant (HH103Ω*NopC*), NopT single mutant (HH103Ω*NopT*), NopL single mutant (HH103Ω*NopL*), and NopC, NopT, and NopL triple mutant (HH103Ω*NopT&NopC&NopL*), the numbers of nodules and dry weights of nodules were evaluated. Soybean was grown in B&D medium (Morad nutrient solution) for 30 days. At least 10 plants with respective phenotypes were considered for the evaluation. Bar, 1 cm. ANOVA was conducted to determine statistical significance (*p* < 0.05, n = 10). A group with the same letter between two groups indicates no significant difference, and a group without the same letter has a significant difference; conversely, distinct marker letters indicate significant differences.

**Figure 3 ijms-24-16521-f003:**
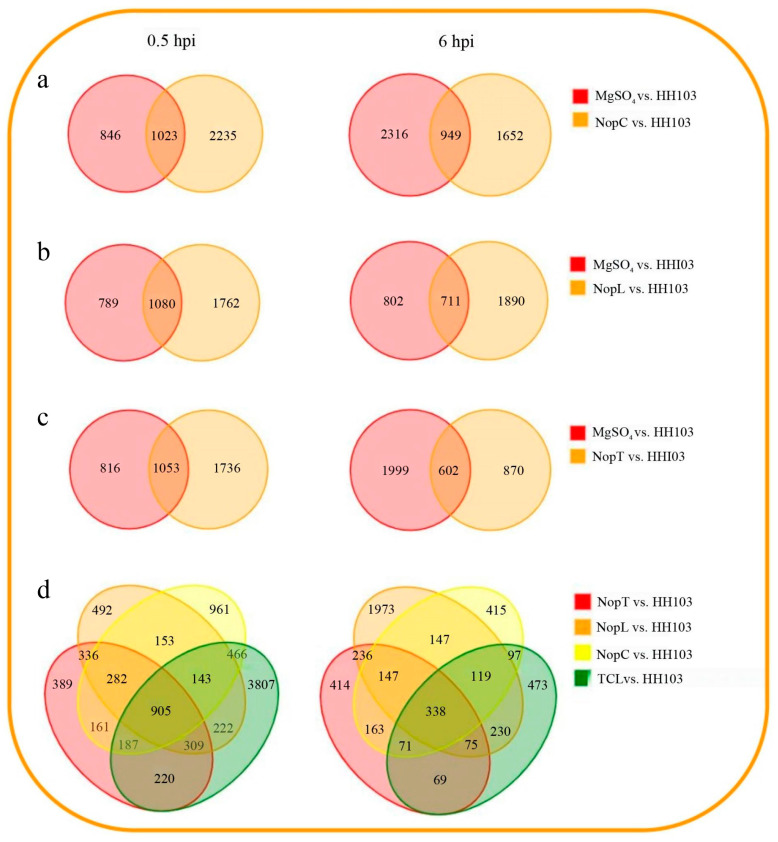
Venn diagram of DEGs. (**a**–**c**) Inoculating NopC, NopT, and NopL with differentially expressed genes compared to wild-type HH103. (**d**) Differential genes in (**a**–**c**) and those inoculated with TCL mutant strains and HH103.

**Figure 4 ijms-24-16521-f004:**
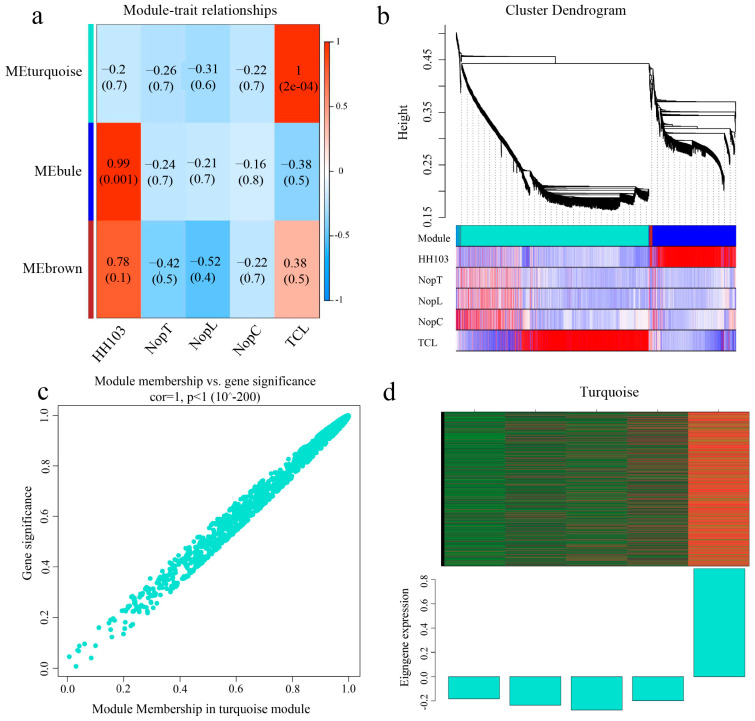
WGCNA analysis of 0.5 hpi DEGs. (**a**) Correlation analysis: the darker the color, the higher the correlation. (**b**) Component analysis of the modules corresponding to the genes in WGCNA. (**c**) Data situation of the turquoise module. (**d**) Global expression heatmap of turquoise module genes.

**Figure 5 ijms-24-16521-f005:**
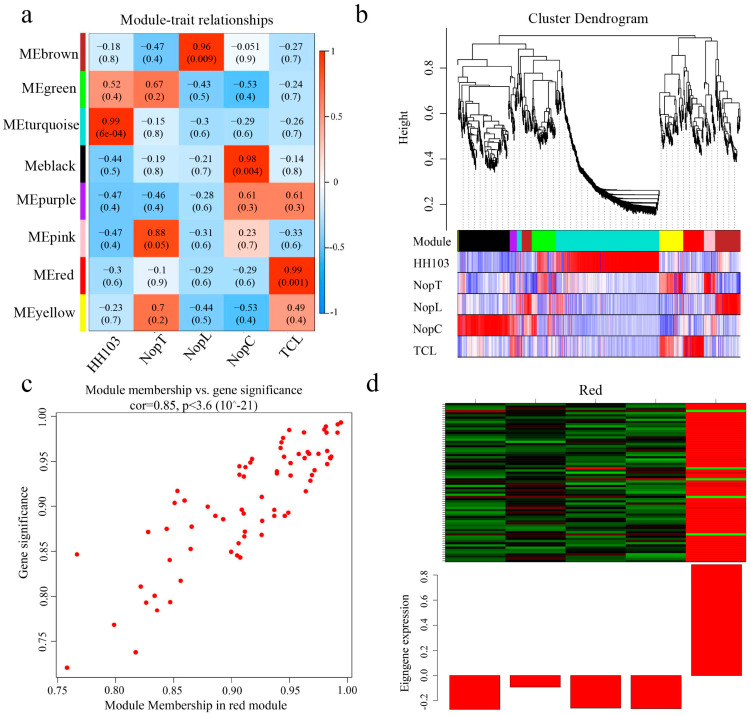
WGCNA analysis of 6 hpi DEGs. (**a**) Correlation analysis: the darker the color, the higher the correlation. (**b**) Component analysis of the modules corresponding to the genes in WGCNA. (**c**) Data situation of the red module. (**d**) Global expression heatmap of red module genes.

**Figure 6 ijms-24-16521-f006:**
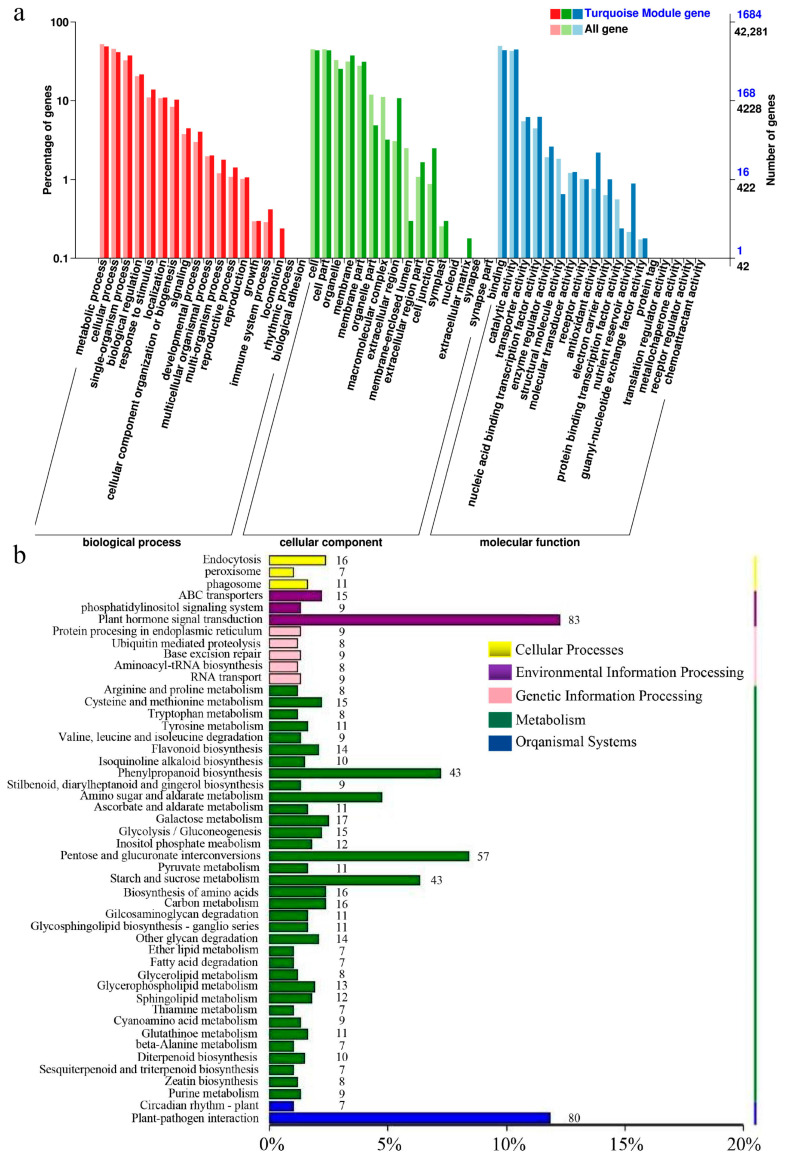
Screening of the hub genes in WGCNA. (**a**,**b**) GO and KEGG enrichment analysis at 0.5 h.

**Figure 7 ijms-24-16521-f007:**
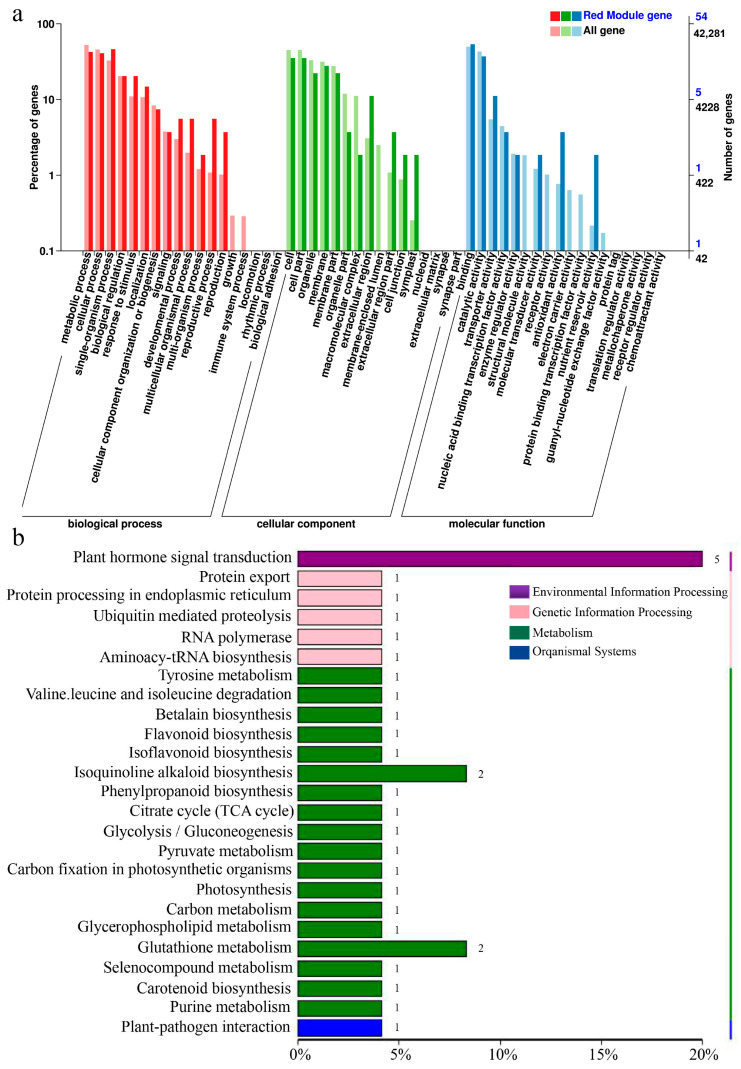
Screening of the hub genes in WGCNA. (**a**,**b**) GO and KEGG enrichment analysis at 6 h.

**Figure 8 ijms-24-16521-f008:**
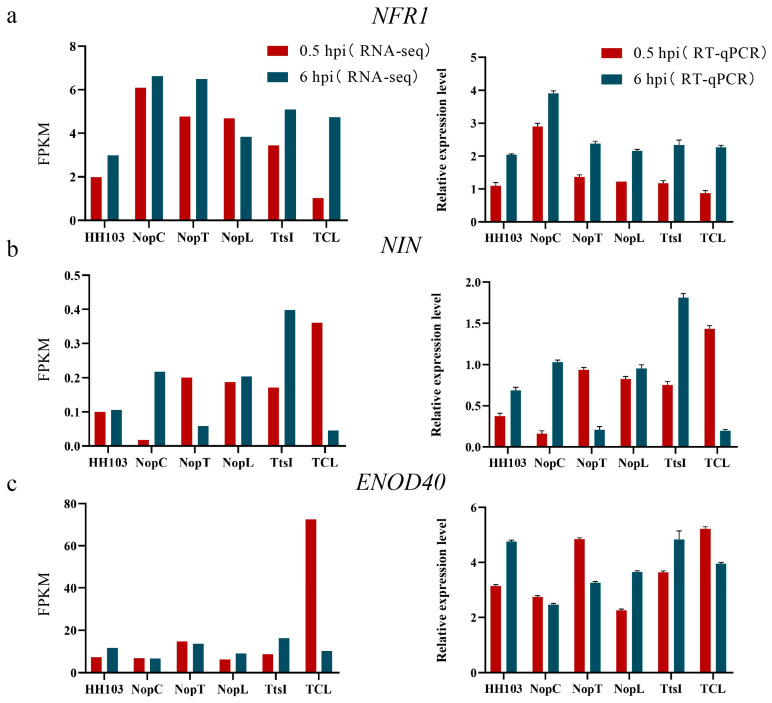
RT-qPCR results of marker genes and *GmPHT1-4* was developed. (**a**–**c**) RNA-seq data and RT-qPCR data of marker genes *NFR1*, *NIN* and *ENOD40* influencing root nodule development at 0.5 and 6 h.

**Figure 9 ijms-24-16521-f009:**
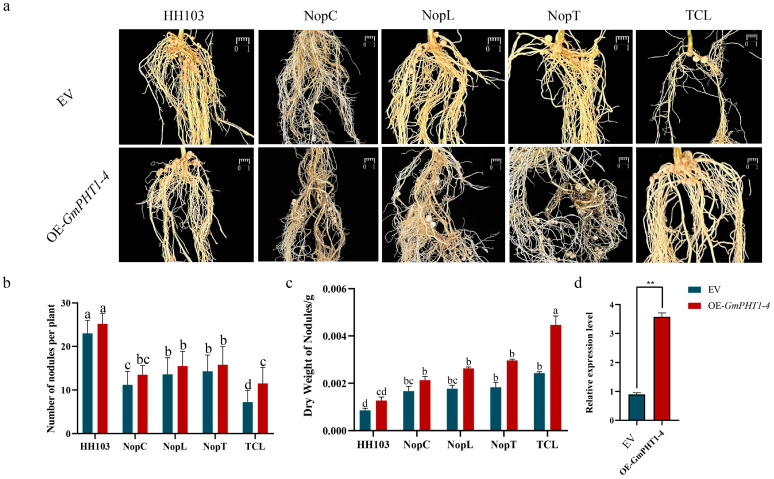
Nodulation phenotype of *Rhizobium* inoculated with soybean e-*GmPHT1-4*. (**a**) Control group (EV group) and experimental group (OE-*GmPHT1-4* group) were inoculated with wild-type *Rhizobium* HH103, and nodule phenotypes of *Rhizobium* mutants NopC, NopT, NopL, and TCL. (**b**) Nodule count statistics. (**c**) Dry weight of nodule statistics. Dry weight statistics were performed for nodules of similar size on each soybean plant to calculate the average weight of each nodule. (**d**) RT-qPCR analysis of OE-*GmPHT1-4* plants. Bar, 1 cm. The numbers of nodules were compared by one-way ANOVA test. A group with the same letter between two groups indicates no significant difference, and a group without the same letter has a significant difference, n = 20. The expression of *GmPHT1-4* was analyzed by *t* test (** < 0.01).

**Figure 10 ijms-24-16521-f010:**
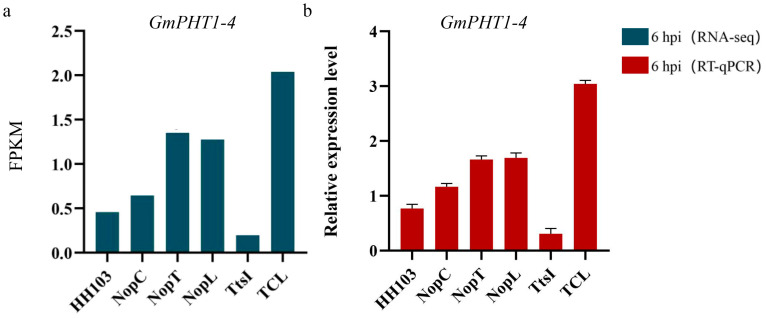
Transcriptome and RT-qPCR results of *GmPHT1-4.* (**a**) Transcriptome results. (**b**) RT-qPCR results of soybean plants 6 hpi with *Rhizobium* HH103, NopC, NopT, NopL, TtsI, and TCL.

**Figure 11 ijms-24-16521-f011:**
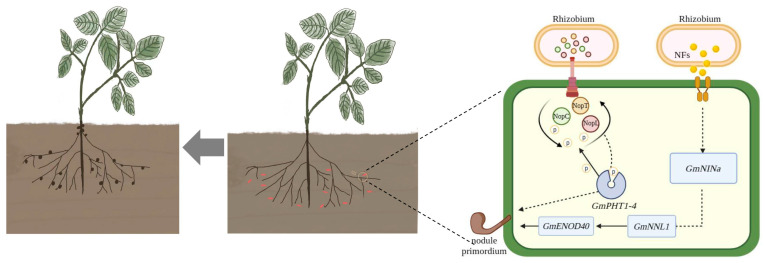
NopC, NopT, and NopL crosstalk model diagram.

**Table 1 ijms-24-16521-t001:** Candidate gene annotation table.

GeneID	kME	Functional Annotation	Expression Position
*Glyma.02G259300*	0.989	PEROXIDASE 52	Root
*Glyma.03G185400*	0.982	PECTATE LYASE 12-RELATED	Shoot
*Glyma.06G174700*	0.977	probable pectate lyase 5	Shoot
*Glyma.07G203100*	0.973	rac-like GTP-binding protein RAC13	Stem
*Glyma.04G011900*	0.956	glucose-1-phosphate adenylyltransferase large subunit 1	Nodules, Leaf, Stem, Shoot, Seed
*Glyma.09G283900*	0.949	protein of unknown function	Seed
*Glyma.12G035700*	0.921	ARG7 auxin-responsive family protein	Leaf
*Glyma.10G247200*	0.912	protein of unknown function (DUF1677) (DUF1677)	Nodules
*Glyma.15G100400*	0.908	calcofluor white hypersensitive protein precursor	Nodules
*Glyma.08G038400*	0.888	transcription repressor OFP8-like	Shoot
*Glyma.07G217900*	0.882	auxin efflux carrier component 3a	Leaf
*Glyma.08G321400*	0.882	aspartyl protease family protein At5g10770	Leaf
*Glyma.12G116200*	0.863	leucine-rich repeat extensin-like protein 4	Root
*Glyma.08G042600*	0.839	plasmodesmata-located protein 6	Shoot
*Glyma.06G145300*	0.836	peroxidase 52-like	Root, Nodules
*Glyma.10G036800*	0.836	inorganic phosphate transporter 1–4	Nodules

(The kME value represents the correlation with TCL, and the larger the value, the higher the correlation; expression position indicates the part of the plant in which the gene is highly expressed, with data obtained from Phytozome).

## Data Availability

Data are contained within the article and Appendix A.

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
