# Peer review of "NopC/T/L Signal Crosstalk Gene *GmPHT1-4"

_ijms, 2023, doi:10.3390/ijms242216521_

Round 1

Reviewer 1 Report

Comments and Suggestions for Authors

See attached review.

Comments on the Quality of English Language

See attached review.

Author Response

Response to Reviewer 1 Comments

Dear Editor,

Thank you pay attention on our manuscript (ijms-2674064). Now, we have completed the modification of the manuscript follow the comments of the reviewers. We prefer to upload the revised manuscript again. We have had the grammar professionally revised. We improved the quality of the picture, made it clearer, and referenced previous research on signal crosstalk. During the modification, we learned a lot of experiences to manuscript preparation. Thank you and the reviewer’s comments again.

If you need other information, please just inform me.

Thank you again!

Sincerely,

Dawei Xin

Answer to reviewer:

Authors should seek the assistance of a naLve English-speaking colleague to help revise manuscript for grammar, punctuaLon, spacing, flow, redundancy, and arLcle use prior to resubmission to the journal. In some places, the wriLng is unclear and makes understanding the experiment difficult.

Thanks for your suggestion, We asked a friend who is proficient in English to revise our manuscript.

In general, more details should be provided in the Materials & Methods secLon. It appears that a logical series of steps was undertaken, there's just not enough informaLon in places for a full determinaLon. I have aZempted to ask for some of these details, but it's too much to fully outline in this review.

Thanks for your suggestion, we have modified the material method to provide more details.

Some experimental details are also included in figure legends and not in the Materials & Methods itself, which generally not preferred.

Thanks for your suggestion, we have modified the material method.

In Figure 2, the figure panels for NopT and TCL are idenLcal images. It is not clear if this is intenLonal or an error, but it must be fixed prior to resubmission of the paper.

Thanks for your suggestion, In Figure 2, due to our negligence, there was a writing error. We have corrected it and changed the NopT graph to the original one.

Many of the figures are too small to read without enlarging the PDF. I imagine when the paper is printed out, they would be quite small as well. May need to break figures up to make sure they stand alone and make sense, but at the same Lme are larger, higher quality, and the text is visible.

Thanks for your suggestion, We have replaced the pictures in the manuscript and enlarged the text.

Proper controls for the hairy root transformaLon experiment?  Empty expression vector?

Thanks for your suggestion, the EV in Figure 9 is an Empty expression vector.

L111-112 The approach for the triparental cross is unclear. Needs more detail.

Thanks for your suggestion, we have made changes.

L121 What is meant by "should be" in this sentence?

Thanks for your suggestion, we changed “The probe sequence position should be in the junction of NopT and Cm, that is, it contains NopT and Cm gene sequences, and the length is about 200 bp” to “The selected probe covered the junction between the NopT and Cm gene sequences, with an approximate length of 200 base pairs”.

L131 Were seeds sterilized with bleach or actual chlorine?! Need to be more specific here.

Thanks for your suggestion, we sterilize soybean seeds with chlorine gas, which has been supplemented in the material method.

L138-139 What do the authors mean by "draw a staLsLcal chart"? Not clear.

Thanks for your suggestion, we have made changes.

L145 How many roots? How do you "design" biological replicates for an RNA-seq experiment? Not clear.

Thanks for your suggestion. The sampling root is about 0.1g, which we have added in the material method.

L149-150 What sohware version was used?

Thanks for your suggestion, we used TBtools-II v2.003, which we have added in the material method.

L153 Product details should be given in full. Not just "Invitrogen".

Thanks for your suggestion, we have supplemented this in line 427 of the Materials Method.

L153 "chloroform" and "isopropanol" do not need to be capitalized.

Thanks for your suggestion, we have made changes.

L156 What is "cDAN synthesis"?

Thanks for your suggestion, what I'm trying to say is Complementary DNA was synthesized employing HiScript II Reverse Transcriptase, which we have added in the material method.

L164 Was only a single plant used for hairy root transformaLon?

Thanks for your suggestion, There was some misunderstanding in our presentation, so we made a comprehensive revision of “Hairy root transformation and positive soybean root detection”.

L173-174 Is there a citaLon for this previous study? If so, please include.

Thanks for your suggestion, we have already referenced it in the body [1].

L189, 191 Rhizobium is a genus name and should capitalized and italicized. See further instances of this throughout the manuscript.

Thanks for your suggestion, we have replaced the full text.

- Figure 1

1.Along the top of panel "a," there are numbers labeled for each lane. It is menLoned in the legend that "(a-c) Strain screening." What are these strains?

Thanks for your suggestion, We labeled the band size and amplified HH103ΩNopT&NopC&NopL strains with different primers, and finally the bands were all in line with the expected serial number as the target strains.

2.Panel "d" of Figure 1 shows the Southern blot results, but these are rather difficult to see. There is a band at ~3000 bp that is much beZer than the smudge at 2000 bp. I would suggest repeaLng the Southern and obtaining a much clearer autoradiograph. This one is not really suitable for publicaLon.

Thanks for your suggestion, We adjusted the contrast of the Southern blot results to make them clear.

  1. Panels "e" and "f" of Figure 1 are too small to read. Even when the PDF is enlarged, the text/figure is blurry and of low quality. I would suggest enlarging and improving these diagrams.

Thanks for your suggestion, We have zoomed in on the TCL build diagram.

- Figure 2

1.In panel "a" the labels fo the plant growth condiLons diagram are too small to read. Again, if the PDF is enlarged, the font looks blurry. Should improve the size and quality of all of the figure panels as best as possible.

Thank you for your advice. We have enlarged Figure a and placed it in the accompanying figure (Figure S7).

2."Morad" and "B&D" soluLons are menLoned in the figure legend. These are not menLoned in the Materials & Methods secLon.

Thanks for your suggestion, The formula of "B&D" nutrient solution is listed in the Material Method.

3.Both panel "b" and panel "c" are box plots. This needs to be indicated in the legend. Further, the black on blue background of the box for TtsI is very hard to see. Can't make out the median line very easily. Perhaps a lighter shade of blue would work beZer.

Thanks for your suggestion, we have indicated in the legend and modified the box diagram.

4.The authors indicate that "Nodule Count StaLsLcs" are presented in Figure 2, panel "b." Although a "soybean nodulaLon experiment" is menLoned in the Materials & Methods, it is not clear how this experiment was designed. Was it a completely randomized design? Or, some other design? The authors menLoned that ANOVA was performed, but there are no ANOVA tables to support the data of this experiment. The authors also indicate that means comparisons have been done using different leZers, but it is not clear what approach

was used for this.

Thanks for your suggestion, The process of soybean nodulation experiment is described in more detail in the

material method.

5.The root masses and nodules in the subpanels of panel "d" are difficult to see. And, it appears that the same plant/root mass photo is shown for NopT and TCL. It is not clear if this is intenLonal or an error. This must be fixed/addressed prior to resubmission of the paper.

Thanks for your suggestion,We are very sorry that “TCL” and “NopT” have made mistakes due to our negligence. We have modified them and changed them into the original pictures.

6.One has to rely on the legend to find some experimental details, all of which should be outlined in the Materials & Methods.

Thanks for your suggestion, we have added more detail to the material approach.

7.Further, I would suggest that Figure 2 be reconsidered. Panel "a" could be a supplementary figure as well as panel "c." It appears that the important informaLon is reported in panels "b" and "c." But, it would be nice for the interested reader to have access to the experimental set up diagram and nodulaLon photos if they so choose.

Thanks for your suggestion, we modified Figure 2 as a whole, put the original figure "a" in the attached figure, and uploaded the original tumor data in the attached figure (Table S2 ,3).

-Figure 3

1.There is plenty of room to increase the size and quality of this figure. The numbers and text should be fully visible at a normal PDF size.

Thank you for your suggestion, we improved the picture quality.

2.Some informaLon to guide the reader could be included in the figure legend.

Thanks for your advice, we added legend information.

3.The MgSO 4 control treatment for the RNAseq experiment is not included in the Materials & Methods. Or, I completely missed it.

Thank you for your suggestion, the RNAseq experiment was carried out against the background of MgSO4 treatment, which we have added to the description in the materials method.

-Figure 4 & 5

1.The panels in Figure 4 are too small. See above suggesLons. May need to break this up into more than one figure so that each piece is clear. Same situaLon for Figure 5.

This is a very good question, we have modified Figure 45 to add some graphs for increased understanding and to divide it into different graphs.

-Figure 7

1.As before, the design, analysis, and means comparison approach for this experiment should be in the Materials & Methods and not only in the legend, if it's there at all.

Thank you for your suggestion, we have reformulated Figure 5 in the Materials Approach.

2.Further, the root systems are very small, and it is difficult to see differences in nodulaLon between the control group and teh over-expression group. Further, this experiment needs another negaLve control, which would be plants with an empty expression vector?

Thanks for your advice, we improved the image quality, and the empty carrier in Figure 7 is EV group [2, 3].

Reference

  1. Li,D.; Zhu, Z.; Deng, X.; et al. GmPBS1, a Hub Gene Interacting with Rhizobial Type-III Effectors NopT and NopP, Regulates Soybean Nodulation. Agronomy 2023, 13(5): 1242.
  2. Wang,J.; Ma, C.; Ma, S.; et al. Genetic variation in GmCRP contributes to nodulation in soybean (Glycine max). Crop J. 2023, 11(2): 332-344.
  3. Ma,C.; Liu, C.; Yu, Y.; et al. GmTNRP1, associated with rhizobial type‐III effector NopT, regulates nitrogenase activity in the nodules of soybean (Glycine max). Food Energy Secur. 2023, 12(4): e466.

Reviewer 2 Report

Comments and Suggestions for Authors

The manuscript titled "NopC/T/L signal crosstalk gene GmPHT1-4" contains interesting research results for science and agricultural practice. The experiment concerns an economically important crop - soybean, and the possibility of improving symbiosis with the Sinorhizobium fredii bacteria. However, the text of the manuscript requires improvement. I included detailed comments in the original text (pdf). After making corrections, it recommends publishing the article in the International Journal of Molecular Sciences

General comments:
correct author affiliations as required by the journal
write the Latin name of soybean in the Abstract
add to keywords: Sinorhizobium fredii
Write briefly about symbiotic bacteria for soybeans:
Bradyrhizobium japonicum
B. elkanii
Bradyrhizobium diazoefficiens
Sinorhizobium fredii ....etc
In the Introduction, briefly describe how soybeans are commercially inoculated around the world
Correct the aim of the experiment and you can add a research hypothesis
Change chapter numbering e.g. Material and Methods according to the journal requirements (https://www.mdpi.com/journal/ijms/instructions)
carefully check the numbers of tables and figures (appendix too) and references in the text,
specify in the description what variety the experiments were performed on (Suinong 14 (SN14) and DN50 (Dongnong 50),
what was the substrate in the pots?

Add: Supplementary Materials: The following supporting information can be downloaded at: www.mdpi.com/xxx/s1, Figure S1: title; Table S1: title; Video S1: title.

Correct the description under Table 1
Correct the caption under Figure 6
You can add chapter 5 Conclusion (I leave it to the authors' decision) and information on what further research is needed
Correct the reference list in accordance with the journal's requirements

I hope that my comments will help the authors improve the text of the manuscript. Thank you for your cooperation.

Author Response

Response to Reviewer 2 Comments

Dear Editor,

Thank you pay attention on our manuscript (ijms-2674064). Now, we have completed the modification of the manuscript follow the comments of the reviewers. We prefer to upload the revised manuscript again. We have had the grammar professionally revised. We improved the quality of the picture, made it clearer, and referenced previous research on signal crosstalk. During the modification, we learned a lot of experiences to manuscript preparation. Thank you and the reviewer’s comments again.

If you need other information, please just inform me.

Thank you again!

Sincerely,

Dawei Xin

Answer to reviewer:

1 Affiliation 1; e-mail@e-mail.com. Affiliation 2; e-mail@e-mail.com. *Correspondence: e-mail@e-mail.com; Tel.: (optional; include country code; if there are multiple corresponding authors, add author initials)

Thanks for your suggestion, we have made changes.

2 Write your affiliation (institution, university)

Thanks for your suggestion, we have made changes.

3 ... soybean (Glycine max (L.) Merr.) ....

Thanks for your suggestion, we have made changes.

4 Write briefly about symbiotic bacteria for soybeans Bradyrhizobium japonicumB. Elkanii, Bradyrhizobium diazoefficiens, Sinorhizobium fredii....

Thanks for your suggestion, we have added relevant expressions in the preface.

5 add: Sinorhizobium fredii

Thanks for your suggestion, we have made changes.

6 Add publications on soybean inoculation (soybean inoculation is a common practice in many countries)

Thanks for your suggestion, we added the relevant references [1].

7 Improve your research aim. You can add a research hypothesis.

Thanks for your suggestion, we have revised this part and added the experimental hypothesis.

8 Materials and Methods and see: https://www.mdpi.com/journal/ijms/instructions change chapter order or see recent articles.

Thanks for your suggestion, we made adjustments to the position of the material method, see https://www.mdpi.com/journal/ijms/instructions.

9 Supplementary Materials: The following supporting information can be downloaded at: www.mdpi.com/xxx/s1, Figure S1: title; Table S1: title; Video S1: title.

Thanks for your suggestion, we add to this later in the article.

10 OK, describe the differences for the varieties in the text of your work.

Thank you for your advice. DN50 was favored over SN14 for its higher efficacy in genetic transformation experiments.

11 What was the substrate in the pots? Morad nutrient solution ?

Thanks for your suggestion. The upper bowl is filled with vermiculite, and the lower is B&D nutrient solution, which we have supplemented in the material method.

12 what variety ?

Thanks for your suggestion, after inoculation with TCL, the nodule number decreased but the nodule volume increased.

13 (Table 1)  ???

Thanks for your suggestion, we have made changes.

14 Correct it.

Thanks for your suggestion, we have made changes.

15 Discussion

Thanks for your suggestion, we have made changes.

16 Figure

Thanks for your suggestion, we have made changes.

17 Conclusions This section is not mandatory but can be added to the manuscript if the discussion is unusually long or complex.

Thanks for your suggestion, We have revised the discussion section

18 Ok, what further experiments should be conducted. 

This is a very good question, phosphorylated motifs of NopC, NopT and NopL were detected, key action sites were screened, non-phosphorylated modification was simulated, nodule phenotypes were observed, stable genetic plants overexpressed and GmPHT1-4 knocked out were cultivated, and phosphorylated mutants of rhizobia were inoculated.

19 add: All authors have read and agreed to the published version of the manuscript.

Thanks for your advice, we've already added that.

20 Author 1, A.B.; Author 2, C.D. Title of the article. Abbreviated Journal Name Year, Volume, page range.

Thank you for your suggestion, we have made changes to the format of the references

21 Abbreviation.

This is a very good question, we have made changes.

  1. Jabborova,D.; Kannepalli, A.; Davranov, K.; et al. Co-inoculation of rhizobacteria promotes growth, yield, and nutrient contents in soybean and improves soil enzymes and nutrients under drought conditions. S Rep-UK, 2021, 11(1): 22081.